

# ECAsT: a large dataset for conversational search and an evaluation of metric robustness

Haya Al-Thani[1], Bernard J. Jansen[2] and Tamer Elsayed[3]

[1] College of Science and Engineering, Hamad Bin Khalifa University, Doha, Qatar
[2] Qatar Computing Research Institute, Hamad Bin Khalifa University, Doha, Qatar
[3] Computer Science and Engineering Department, Qatar University, Doha, Qatar

## ABSTRACT

The Text REtrieval Conference Conversational assistance track (CAsT) is an annual conversational passage retrieval challenge to create a large-scale open-domain conversational search benchmarking. However, as of yet, the datasets used are small, with just more than 1,000 turns and 100 conversation topics. In the first part of this research, we address the dataset limitation by building a much larger novel multi-turn conversation dataset for conversation search benchmarking called Expanded-CAsT (ECAsT). ECAsT is built using a multi-stage solution that uses a combination of conversational query reformulation and neural paraphrasing and also includes a new model to create multi-turn paraphrases. The meaning and diversity of paraphrases are evaluated with human and automatic evaluation. Using this methodology, we produce and release to the research community a conversational search dataset that is 665% more extensive in terms of size and language diversity than is available at the time of this study, with more than 9,200 turns. The augmented dataset not only provides more data but also more language diversity to improve conversational search neural model training and testing. In the second part of the research, we use ECAsT to assess the robustness of traditional metrics for conversational evaluation used in CAsT and identify its bias toward language diversity. Results show the benefits of adding language diversity for improving the collection of pooled passages and reducing evaluation bias. We found that introducing language diversity via paraphrases returned up to 24% new passages compared to only 2% using CAsT baseline.

# INTRODUCTION

Conversational search (CS) has gained more interest due to the popularity of conversational agents like Amazon's Alexa and Apple's Siri. The conversational mode is increasingly becoming a standard mode of interaction for search due to the increasing number of devices often used on the move without a keyboard (*Culpepper, Diaz & Smucker, 2018*). Search trends show that users many times prefer conversational forms of search. As of December 2020, most Google search sessions (64.8%) did not end with a click but with short, concise answers (*Fishkin, 2020*). Despite the exponential progress of digital assistants and their speech interfaces, they still struggle as useful exploratory search tools. A major

Corresponding author
Haya Al-Thani,
hayaalthani@hbku.edu.qa

**Table 1  TREC CAsT sample topic from the 2020 dataset.**

| | |
|---|---|
| **T1:** | **What are some interesting facts about bees?** |
| R1: | Fun facts about bees…Honey never spoils. |
| **T2:** | **Why doesn't it spoil?** |
| R2: | The water content …support microbial growth. |
| **T3:** | **Why are so many dying?** |
| R3: | Honeybees are dying…industry in America itself. |

obstacle to building conversational systems is the lack of large datasets to create effective and efficient CS systems. We address this limitation by creating a larger and more diverse dataset that can be used to train and test CS systems or evaluate performance metrics in CS.

CS introduces a variety of under-explored challenges compared to traditional search. Traditional search is conducted with a well-formed query where a system returns a ranked results list. However, CS often relies on iterated questions or "turns" between the user and the CS system. An example of such an interaction can be seen in Table 1. The user starts a conversation with the first turn "T1", fully stating their information need. The system then responds with "R1", prompting the user to ask a follow-up turn related (or not) to their initial information need. Subsequent turns, such as "T2", often contain omissions and references to missing context only found in previous turns or responses. Building a CS system that correctly follows dialogue context is a foremost challenge.

The Text REtrieval Conference Conversational Assistance Track (TREC CAsT) (*Dalton, Xiong & Callan, 2019*) is an annual conversational passage retrieval challenge to create a large-scale benchmark for open-domain CS. The CAsT dataset contains open-domain multi-turn conversations and responses. A *multi-turn* conversation is comprised of multiple questions, where each question is related to others in the same conversation. *Single-turn* questions, on the other hand, are usually self-contained questions that are unrelated to others in the dataset. Table 1 is an example of the first three turns and responses of topic 83 from the CAsT 2020 dataset. The main challenge in CAsT is maintaining context-awareness throughout the conversation to retrieve a list of relevant passages. *Context-awareness* is when missing information need is resolved at every turn in the conversation.

In CS, features such as efficiency, effectiveness, and reliability greatly impact user experience (*Guichard et al., 2019*). If a conversational agent cannot interpret users' requests, users are unlikely to use the agent repeatedly, especially not for complex searches. Building a CS system that can understand diverse natural languages and accurately measure system understanding of user information needs is essential. One of the challenges is the lack of large conversational datasets with enough language diversity to build such effective, efficient, and robust systems. The largest CAsT dataset, as of this writing, contains only 131 topics with a total of 1,203 turns (*Dalton, Xiong & Callan, 2021*). This is very small compared to other IR datasets, such as MS MARCO, which has over three million questions (*Nguyen et al., 2016*).

**Table 2  ECAsT paraphrase example of topic 83 turn 2 from CAsT 2020.**

| Context-independent turns | Context-dependent turns |
|---|---|
| **Original CAsT turns:** | |
| Why doesn't honey spoil? | Why doesn't it spoil? |
| **Paraphrases:** | |
| Why does honey never rot? | Why does it never rot? |
| Is honey spoiling? | Is it spoiling? |
| Does honey spoil? | Does it spoil? |
| Why doesn't honey rot? | Why doesn't it rot? |
| Why is honey non-perishable? | Why is it non-perishable? |
| Does honey go bad or expire? | Does it go bad or expire? |
| Why does honey never expire? | Why does it never expire? |

In this research, we create a novel conversation paraphrases dataset that is both larger and more diverse than existing datasets. We make the dataset, called Expanded-CAsT (ECAsT), publicly available for the advancement of CS research. Here, we present how ECAsT is built from only the TREC CAsT datasets as a resource using both neural models and human-in-the-loop diversification. ECAsT is constructed with CAsT turns using automatic paraphrase generation and commercial search engine tools, such as the search engine results pages and the "People Also Ask" feature in the search engine. The ECAsT dataset significantly augments the CAsT dataset by more than 665%. Table 2 shows an example of a paraphrased CAsT turn in ECAsT.

We also use the newly created paraphrase dataset to test the robustness of CS evaluation to language diversity. After introducing language diversity, we identify weaknesses inherent in CS evaluation, and we suggest solutions for making evaluation metrics more robust.

We further use the ECAsT dataset to investigate the robustness of CAsT evaluation. Many IR benchmarks, such as TREC CAsT, evaluate the effectiveness of IR systems on a limited set of topics using their corresponding relevance judgments. Relevance judgments are assessed by TREC using standard pooling, where passages are judged according to their relevance to a turn. All unjudged passages are considered not relevant. Each turn or information need in CAsT is expressed by a single query. However, research has shown that users often express a single information need in a variety of ways (*Zuccon, Palotti & Hanbury, 2016*; *Bailey et al., 2017*). The TREC evaluation method is inherently biased against systems with different information that need expressions that do not return judged passages (*Büttcher et al., 2007*). This misrepresents the quality of systems and would lead to incorrect conclusions about their performance.

CS systems trained to handle language diversity would naturally be penalized during evaluation for returning passages not included in the relevance judgment pool, regardless of the actual relevance of the passage. We examine the magnitude of this bias in TREC CAsT by studying the robustness of CAsT evaluation metrics to paraphrasing. By creating linguistically diverse conversations, we found that the primary metric used by CAsT (NDCG@3) is volatile with paraphrases. We found that the drop in NDCG@3 does not accurately reflect the quality of returned passages, but it is due to incomplete relevance

judgments. We conclude that by including paraphrases in the pooling process, evaluation metrics will be more robust and accurately reflect how systems handle language diversity.

This research aims to answer the following research questions:

- **RQ1:** *How can we employ paraphrasing to augment multi-turn conversation datasets using multiple neural models?*
- **RQ2:** *How well can we paraphrase context-dependent turns compared to context-independent turns?*
- **RQ3:** *Can automatic paraphrase generation and human-in-the-loop improve paraphrase quality and diversity?*
- **RQ4:** *Is the TREC CAsT evaluation metric sensitive to language variation via paraphrasing? How is the metric affected by incomplete relevance judgments?*

In summary, our contributions are:

- We create and release ECAsT, a novel multi-turn conversation paraphrase dataset with 9,214 turns.
- We show through utilizing the proposed multi-stage paraphrasing solution that we can paraphrase context-dependent turns just as well as traditional single-turn paraphrasing using context-independent turns.
- We combine automatic paraphrase generation and human-in-the-loop solutions to create high-quality diverse conversation paraphrases from the original CAsT datasets that can be used in many applications.
- We take a critical look at the TREC CAsT evaluation methods and their robustness to paraphrases using the newly created ECAsT dataset. We conclude that introducing language variation *via* paraphrases increases the diversity of returned passages assessed in the pooling method. This makes evaluation more robust to language diversity.

The article is organized into the following sections: the next section details a comprehensive literature review of related fields. After that, we present the methodology and different stages of the solution. In the following section, the experimental setup is explained. We then present the evaluation results, followed by discussion and implications. The final section is the conclusions and future works.

# LITERATURE REVIEW

In this section, related work will be reviewed from two perspectives: conversational search systems and paraphrasing systems.

## Conversational search systems

Conversational search (CS) is applied in many fields such as recommendation systems, e-health and personality recognition (*Aliannejadi et al., 2020*; *Velicia-Martin et al., 2021*; *Shen et al., 2023*). Deep learning solutions for CS have replaced more traditional rule-based approaches (*Onal et al., 2018*; *Gao, Galley & Li, 2018*; *Li et al., 2022*). A main challenge is how to ensure conversational context-awareness (*Vtyurina et al., 2017*). Conversational context is essential for understanding user intent, abusive language classification, and

many other applications (*Aliannejadi et al., 2019*; *Ashraf, Zubiaga & Gelbukh, 2021*; *Liu et al., 2022*). Conversational query reformulation (CQR) uses pre-trained sequence-to-sequence models to resolve context in ambiguous turns. *Elgohary, Peskov & Boyd-Graber (2019)* fine-tune the text-to-text transfer transformer (T5) model (*Raffel et al., 2020*) to take a conversation's entire history, along with the turn to be resolved, and output a context-independent turn.

TREC CAsT aims to establish a large-scale open-domain CS benchmark using a conversational passage retrieval challenge. CAsT organizers release an evaluation test collection annually where participants are asked to return a list of ranked passages that answer each turn in the conversation collection (*Dalton, Xiong & Callan, 2021*). Conversations are built based on real user information needs from Bing search sessions (*Rosset et al., 2020*). Organizers manually review and filter conversations to make sure they are meaningful and rewrite them to make them conversational. The dataset is composed of "raw" context-dependent turns, and "manual" context-independent turns. Retrieval using raw turns is more challenging due to their lack of context. CAsT is currently in its fourth year.

CS evaluation and how to gauge system effectiveness has received wide debate in the literature (*Anand et al., 2020*). Many IR evaluation measures are derived from recall and precision (*Buckley & Voorhees, 2004*). These approaches are used for offline evaluation of CS using test collections with relevance judgements. Other measures are based on user interaction models, and neural models that score user satisfaction and interaction (*Lipani, Carterette & Yilmaz, 2021*). Modern evaluation solutions train models to generate metrics, such as usefulness, to measure user behaviour and estimate satisfaction (*Rosset et al., 2020*).

TREC CAsT evaluation is based on the pooling method, where organizers pool passages from different participant solutions and manually label them according to their relevance. Scored passages are added to a relevance judgment file called QREL. If a system retrieves a passage not included in the QREL, it is counted as not relevant. This makes the CAsT evaluation biased against systems that return unjudged passages and leads to possibly incorrect conclusions about the quality of the system under investigation (*Voorhees, 2001*). To overcome incomplete relevance judgments, measures such as bpref and infAP, were proposed and later included in the official TREC evaluation tool (*Büttcher et al., 2007*; *Yilmaz & Aslam, 2006*). Bpref ignores unjudged passages and considers relevant documents not included in the ranking. Inferred AP (infAP) estimates current precision when encountering unjudged passages in the ranking. *Clarke, Vtyurina & Smucker (2021)* discuss the limitation of CAsT evaluation and propose measuring performance by comparing passage similarity to a preferred ordered list based on re-ordering top relevant passages.

## Paraphrasing systems

Paraphrasing input sentences or questions is the process of expressing the same meaning or information need using different words and expressions and is a valuable resource for developing experimental CS systems (*Kauchak & Barzilay, 2006*). Paraphrasing is a data augmentation technique that increases the size of available labeled data by creating synthetic data while preserving original class labels (*Feng et al., 2021*). Data augmentation

has gained a lot of interest in the natural language processing (NLP) community due to the increase of studies in low-resource domains, new tasks, and the popularity of large-scale neural models that need large amounts of training data (*Feng et al., 2021*). Most data augmentation techniques rely on word-level or synonym-based substitutions (*Wang & Yang, 2015*; *Kobayashi, 2018*). According to *Barzilay & McKeown (2001)*, there are three main approaches to the paraphrasing problem: manual collection, using existing lexical resources and corpus-based extraction.

Manual collection of paraphrases is usually performed using human annotators on crowd-sourcing platforms such as Amazon's Mechanical Turk. This technique is used by *Chklovski (2005)* where paraphrases are collected *via* a game where users must reformulate a given sentence based on hints. To collect more diverse paraphrases, *Yaghoub-Zadeh-Fard et al. (2020)* were inspired by another game called Taboo and gave workers a list of taboo words they were not allowed to include in their paraphrases.

The second approach uses lexical resources such as substitution of words (*Guichard et al., 2019*), or making syntactical changes to the original sentence (*Iyyer et al., 2018*). *Hassan et al. (2007)* incorporate lexical, semantic, and probabilistic methods to find the most likely substitute for a word given a context.

Corpus-based extraction is the most common, where paraphrases are collected from texts such as news articles or translation books. In the work of *Quirk, Brockett & Dolan (2004)*, a large number of sentence-pairs were collected from newspapers to train a statistical translation tool. If two English phrases are translated into the same foreign phrase, they can be considered paraphrases of each other (*Ganitkevitch, Van Durme & Callison-Burch, 2013*).

Deep learning is being applied to paraphrasing with great success. *Prakash et al. (2016)* employ a stacked residual Long Short-Term Memories (LSTM) network to enlarge paraphrasing model capacity. *Gupta et al. (2018)* proposed combining generative models based on variational auto-encoders with sequence-to-sequence models based on LSTM to generate paraphrases. Pre-trained language transformer models have outperformed previous works in many NLP tasks (*Devlin et al., 2019*; *Raffel et al., 2020*; *Radford et al., 2019*). These models' generative capabilities can be leveraged to produce high-quality paraphrases (*Ponkiya et al., 2020*).

Question paraphrase generation, where the goal is to generate a paraphrase for a given question, has played an important role in understanding NLP systems (*Zhou & Bhat, 2021*). Question paraphrases are helpful in evaluating an agent's understanding ability and the ability to interpret diverse language expressions. *Duboue & Chu-Carroll (2006)* found that using paraphrases on a state-of-art question answering system could increase the original question's potential performance. *Gan & Ng (2019)* used paraphrases to create an adversarial test set that uses context words close to incorrect answers in order to confuse the system. Including human-in-the-loop elements introduces more diversity to stress test models *via* adversarial data (*Wallace et al., 2019*). *Penha, Câmara & Hauff (2022)* used paraphrasing to test evaluation robustness to language variation. However, no existing research has been done on paraphrase generation for turns that depend on context not included in the input question. Question paraphrasing research is mainly focused on

single-turn questions. To date, as of this research, even conversational question paraphrases have a focus on single-turn conversations (*Guichard et al., 2019*; *Kacupaj et al., 2021*).

### Position of our study

The literature review of previous CS and paraphrasing research demonstrate the different complexities present in this field. Understanding user information needs is essential to creating a system that can adapt to different languages and user expressions. To the best of our knowledge, no existing work has been done to explore multi-turn paraphrase generation and its use to test the robustness of multi-turn conversational search. We explore how using a multi-stage solution can paraphrase turns without context. Combining automatic paraphrase generation with human-in-the-loop techniques can improve paraphrase quality and diversity using human ingenuity. The goal is to combine these approaches to create a diverse multi-turn conversation paraphrase dataset to assess the robustness of CAsT evaluation and its bias, if any, towards language diversity. This allows us to detect potential weaknesses and limitations in the evaluation scheme to suggest improvements to the current CS evaluation.

## METHODOLOGY

In this section, we present how ECAsT is created and how we use the conversation paraphrase dataset to assess the robustness of CS evaluation. First, we present the different elements and complexities in the CAsT dataset and the elements we aim to have in the ECAsT dataset. The approach is then detailed by explaining each stage of building the new dataset and how it is used to test the robustness of evaluation to language diversity.

**Dataset**    In the CAsT dataset, each conversation comprises a series of $N$ raw turns $\{u_1, u_2, u_3, \ldots, u_N\}$. According to CAsT terminology, *raw turns $u_i$* are context-dependent, while *manual turns $m_i$* are context-independent. *Context-dependent* turns are turns that contain omissions and co-references while *Context-independent* turns are self-contained turns that clearly express the user's information need without omissions and references. Similarly, context-dependent paraphrases will be called *raw paraphrases $pr_i$* and context-independent paraphrases are *manual paraphrases $pm_i$*. The goal is to have a set of raw paraphrases $Pr_i$ for each raw turn $u_i$ in CAsT such that $Pr_i = \{pr_i^1, pr_i^2, \ldots, pr_i^l\}$, where each $pr_i^l$ is a unique raw paraphrase. To create a complete paraphrase collection for each conversation, we should also have a set of manual paraphrases $Pm_i$ for each manual turn $m_i$ such that $Pm_i = \{pm_i^1, pm_i^2, \ldots, pm_i^l\}$, where $pm_i^l$ is a unique manual paraphrase of $m_i$.

CAsT releases an evaluation set each year, as of this writing. CAsT 2019, 2020, and 2021 will be used to build the novel ECAsT dataset. These datasets will be referred to as $CAsT^{year}$, where "year" denotes the year the set was released. Table 3 lists the various notations used in this solution.

**Approach**    To augment the size of the CAsT dataset *via* paraphrasing, turns must go through multiple stages. CAsT turns are automatically paraphrased using a pre-trained transformer-based multi-stage solution followed by human-in-the-loop techniques using search engine results page and the "People Also Asked" feature. The different stages of the solution are illustrated in Fig. 1. Stages 1 to 4 describe how ECAsT is built using TREC

**Table 3 Notations used in the solution.**

| Name | Description |
|---|---|
| $u_i$ | Raw conversation utterance at turn $i$. Raw turns are context-dependent. |
| $m_i$ | Manual conversation utterance at turn $i$. Manual turns are context-independent. |
| $Pr_i$ | Set of raw paraphrases for turn $u_i$. Raw paraphrases are context-dependent. |
| $pr_i^l$ | Single unique raw paraphrase for turn $u_i$. |
| $Pm_i$ | Set of manual paraphrases for turn $u_i$. Manual paraphrases are context-independent. |
| $pm_i^l$ | Single unique manual paraphrase for turn $u_i$. |
| $r_i$ | Reformulated utterance at turn $i$ rewritten by the trained model. |
| $h_i$ | Conversation history made up of previous raw turns at turn $i$. |
| $c_i$ | Canonical response for turn $u_i$. |
| $CAsT^{year}$ | CAsT dataset release, where $year$ is either 2019, 2020, or 2021. |
| $ECAsT$ | Expanded-CAsT dataset that augments CAsT data using paraphrasing. |

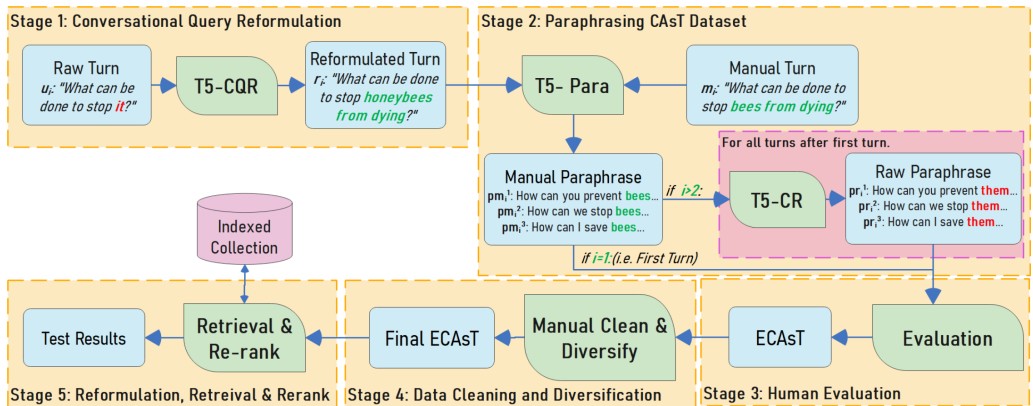

**Figure 1 Multi-turn conversation reformulation and paraphrasing pipeline.**

CAsT as a resource. In Stage 5, we use ECAsT to test how CS evaluation reacts to the introduction of paraphrases. The stages are:

- **Stage 1: Conversational Query Reformulation** The aim of this stage is to reintroduce context into raw turns so they can be used as input to a neural paraphrase generation model.
- **Stage 2: Paraphrasing CAsT Dataset** The aim of this stage is to generate manual and raw paraphrases using reformulated and manual turns in CAsT.
- **Stage 3: Human Evaluation** This stage aims to evaluate generated paraphrases through crowd-sourcing.
- **Stage 4: Data Cleaning and Diversification** The aim here is to manually review, clean, and diversify the new paraphrases with lexical substitutions using a human-in-the-loop approach exploiting commercial search engine tools.
- **Stage 5: Reformulation, Retrieval, and Re-Ranking** In this stage, the new ECAsT dataset is used to evaluate the robustness of CAsT evaluation by using the paraphrases for retrieval and re-ranking.

**Table 4  Example of paraphrases generated without reformulation using turns from CAsT 2020.**

| Topic 83 turn 3 (Context: Bees) | Topic 96 turn 6 (Context: Snowboarding) |
|---|---|
| **Original CAsT Turns:** | |
| Why are so many dying? | How can I teach myself to get started? |
| **Paraphrases:** | |
| Why are many people dying each year? | How can I start teaching myself to be more successful in life? |
| Why do so many people die? | Tell me the best way for learning? |
| Why do people die? | What should I teach myself to start? |
| Why are so many people dying? | How do I teach myself to start my own business? |

## Stage 1: Conversational query reformulation

The goal of this stage is to rewrite raw turn $u_i$ into a context-independent turn $r_i$. Conversational query reformulation is an essential step before paraphrasing. Current paraphrasing models are trained using single-turn questions. Using these models on CAsT raw turns directly would often result in the addition of incorrect context leading to noisy paraphrases. Table 4 has an example of two raw turns and their paraphrases. For both turns, the model adds irrelevant context which changes the information need(s) of the turn making the paraphrases incorrect.

To reformulate the raw turn, the T5-CQR model is used. T5 is a powerful generative model that is computationally expensive to train but produces high-quality reformulations of turns. T5 is fine-tuned using the CANARD dataset (*Elgohary, Peskov & Boyd-Graber, 2019*). The CANARD dataset is pre-processed for training by concatenating the raw conversation history at turn $i$, $h_i = \{u_1, u_2, \ldots, u_{i-1}\}$, with the raw turn $u_i$ and using the manual turn $m_i$ as the model output. Similarly, CAsT raw turn $u_i$ is reformulated into a context-independent turn $r_i$ using its conversation history $h_i$. After reintroducing context back into raw turns, reformulated turns can be paraphrased with the correct information need (Fig. 2 illustrates how the same turn is paraphrased with and without T5-CQR).

## Stage 2: Paraphrasing CAsT dataset

The aim of this stage is to generate raw and manual paraphrases to augment CAsT data by building a novel conversation paraphrase dataset. Neural models are used to generate the paraphrases.

To implement the paraphrase generation model T5-Para, we fine-tune the T5 model using the Quora Question Pairs (QQP) dataset (*Iyer, Dandekar & Csernai, 2017*). QQP is one of the most popular existing datasets for question paraphrases. It contains pairs of questions that are labeled as duplicates or unique. Duplicates are questions with the same information need but different expressions, while unique questions have a different information need. To train T5-Para, we are only considering pairs labeled as duplicates, which is around 150 k pairs.

Figure 2 illustrates how T5-Para performs with raw *versus* reformulated turns. In Fig. 2A, T5-Para is used on raw turns resulting in incorrect paraphrases. T5-Para can not handle missing context due to how it is trained. We can see in Fig. 2B, adding a step to reformulate raw turn results in correct paraphrases due to the reintroduction of context with T5-CQR.

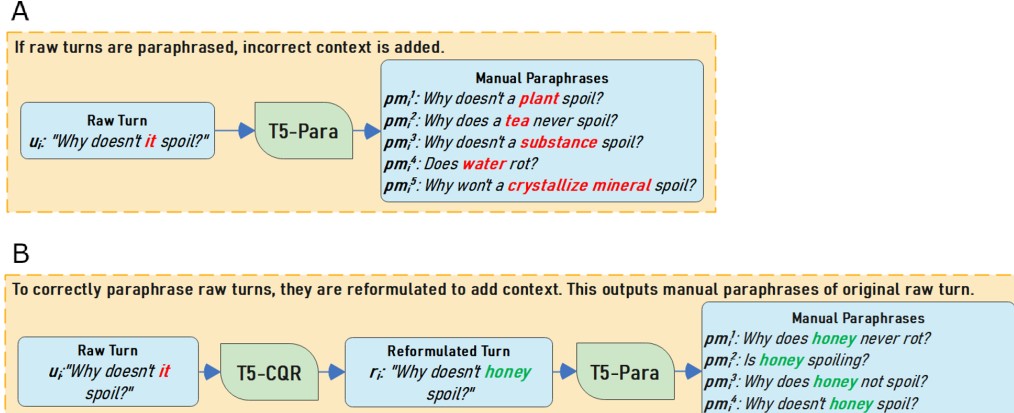

**Figure 2** (A–B) Paraphrasing CAsT examples to illustrate the benefits of reformulation before para-phrasing.

Using T5-Para, we input source turns using both the reformulated turn from stage 1, $r_i$, and its corresponding manual turn in CAsT, $m_i$, to get a list of manual paraphrases $Pm_i$. Using these two sources for paraphrasing not only generates the largest number of paraphrases, but it also allows us to investigate the quality of paraphrases generated by automatically reformulated and manually rewritten turns. Using $m_i$ also guarantees correct information need since these turns are manually written by CAsT authors, whereas $r_i$ could contain incorrect information need because it is automatically generated using T5-CQR. To generate more than one paraphrase per input, top-k and top-p sampling was used (*Fan, Lewis & Dauphin, 2018*; *Holtzman et al., 2019*). Top-k sampling decreases unreliable tails in the probability distribution of neural models. While top-p makes sure the next word chosen by the model is from the top probable choices.

For turns in $CAsT^{2021}$, the organizers included user feedback to reflect users' dissatisfaction with previous responses by adding statements such as "What? No, I want to know...". Another addition to the dataset is user revealments which reveal extra clues about the user's intent to the system, such as "I live in Seattle and have a big lawn." In this stage of paraphrasing, feedback, and revealment sentences are removed since it is observed that T5-Para performs best with single-question inputs. These were removed by segmenting the turn into separate sentences and keeping only the final question portion of the turn.

After that, all turns after the first at depth 2 and above are taken into another T5 model for context removal, T5-CR. First turns always contain full context. Having both manual and raw paraphrases in the final ECAsT dataset is essential to reflect real-world conversations and allow for interesting context-awareness experiments. T5-CR is fine-tuned using the CANARD dataset (*Elgohary, Peskov & Boyd-Graber, 2019*). The system is trained to receive manual turns and output raw turns with omissions and references. It generally emulates how the turn would be if it appeared in the middle of a conversation. In Fig. 3, we add the final step with T5-CR to generate raw paraphrases.

To create raw paraphrases, T5-CR is used to add references and remove context from manual paraphrases.

Raw Turn
$u_i$:"Why doesn't *it* spoil?"

T5-CQR

Reformulated Turn
$r_i$: "Why doesn't *honey* spoil?"

T5-Para

Manual Paraphrases
$pm_i^1$: Why does *honey* never rot?
$pm_i^2$: Is *honey* spoiling?
$pm_i^3$: Why does *honey* not spoil?
$pm_i^4$: Why doesn't *honey* spoil?
$pm_i^5$: Why does *honey* from bees never rot?

T5-CR

Raw Paraphrases
$pr_i^1$: Why does *it* never rot?
$pr_i^2$: Is *it* spoiling?
$pr_i^3$: Why does *it* not spoil?
$pr_i^4$: Why doesn't *it* spoil?
$pr_i^5$: Why does *it* never rot?

**Figure 3** Removing context using T5-CR from manual paraphrases to get raw paraphrases after T5-CQR reformulation and T5-Para paraphrasing.

## Stage 3: Human evaluation

In this section, we evaluate the quality of the generated paraphrases. After stage 2, we have over 10 k unique paraphrases of the original 1,203 turns in CAsT with an average of 8.4 paraphrases per turn. Most automatic evaluation metrics for text generation mainly focus on n-gram overlaps instead of meaning. That is why human evaluation more accurately measures generated paraphrase quality (*Zhou & Bhat, 2021*). Human evaluation is naturally more expensive compared to automatic evaluation so a sample of the paraphrase collection is used.

Human evaluation was conducted on Amazon Mechanical Turk (mTurk). Amazon mTurk was used instead of recruiting local volunteers due to the size of the evaluation task (12,750 tasks). The evaluation task also does not require any domain-specific knowledge from annotators other than a general proficiency in the English language. Annotators were informed beforehand that the task is part of a research study. HIT approval rate, which represents the percentage of completed tasks approved by the requester, was set to above 98% to enforce higher quality annotators. To make certain annotators were English speakers, the location was set to one of either the United States or the United Kingdom. Incentives equate to the US minimum hourly wage with 30 cents for the on average 2.5 min task. Concerning sampling, 2550 paraphrases were randomly sampled from all three CAsT datasets equally; $CAsT^{2019}$, $CAsT^{2020}$, and $CAsT^{2021}$. We sample paraphrases generated with both reformulated and manual source turns equally as well. The effects of the paraphrase source turn are analyzed in later sections. Five annotators were assigned to each task due to the complexity of the evaluation. Annotators were asked to rate five different measures on a scale of one to five. The human evaluation measures are summarized in Table 5.

CAsT canonical response $c_i$ is used as the relevant passage for measuring passage relevance. Canonical responses are passages in the dataset selected by CAsT organizers to represent relevant responses to turns. For $CAsT^{2019}$, the dataset does not include canonical responses for any turns. To evaluate passage relevance for this dataset we use the available relevance judgment files (QREL). The passages in this file are scored according to the assessment guidelines outlined in Table 6 (*Dalton, Xiong & Callan, 2019*). Passages with a score of *4* (when not available *3*) were used instead of canonical responses. With $CAsT^{2021}$, we have incorrect canonical responses in the dataset. To eliminate those, we rely on available user feedback included in the turns. Sentiment analysis was applied on the turns, and if any negative feedback was found, the canonical response was replaced with a relevant passage extracted from the QREL file as with $CAsT^{2019}$.

**Table 5** An explanation of the different human evaluation measures used to evaluate quality of the generated paraphrases.

| Measure | Definition | Importance |
|---|---|---|
| Semantic similarity | This measures the similarity between CAsT manual turn $m_i$ and manual paraphrase $pm_i^l$ in meaning. | The aim is to ensure information need is retained after T5-Para paraphrasing. |
| Language diversity | This measures the language and/or sentence structure differences between CAsT manual turn $m_i$ and manual paraphrase $pm_i^l$. | The aim is to have as much language diversity between manual turn and paraphrase. |
| Raw semantic similarity | This measures the similarity between manual paraphrase $pm_i^l$ and raw paraphrase $pr_i^l$ in meaning. | The aim is to ensure that information need is retained after T5-CR transformation. |
| Conversational entailment | This measures if manual paraphrase $pm_i^l$ is relevant as part of the overall conversation history $h_i$. | The aim is to make sure new paraphrase is on-topic and fits well with conversation history. |
| Passage relevance | This measures how relevant a passage is to manual paraphrase $pm_i^l$. | The aim is to ensure relevant labels associated with original CAsT turns are conserved after paraphrasing. |

**Table 6** Relevance judgements guidelines.

| | |
|---|---|
| 4- Fully Meets | The passage is a perfect answer to the turn. It focuses only on information need. |
| 3- Highly Meets | The passage answers the turn. It contains limited extraneous information. |
| 2- Moderately Meets | The passage answers the turn partially but focuses on unrelated information. |
| 1- Slightly Meets | The passage includes some relevant content, but doesn't answer the turn directly. |
| 0- Fails to Meet | The passage is not relevant to the turn. |

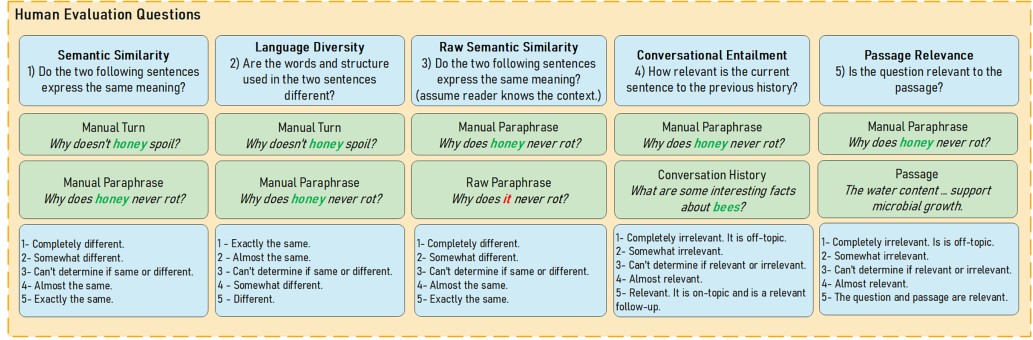

**Figure 4** Paraphrasing evaluation measures, questions, and 5-scale rating presented to annotators during the mTurk evaluation process.

Figure 4 summarizes the five measures, questions asked to annotators, and 5-point rating scale used for the evaluation.

## Stage 4: Data cleaning and diversification

We want to augment the CAsT dataset by creating a conversation paraphrase dataset called ECAsT that is semantically similar while having as much language diversity as possible. The aim of this stage is to manually review the generated paraphrases and to improve their diversity with a human-in-the-loop approach.

**Table 7** Examples of paraphrase diversification.

| Original CAsT topic 83 turn 2 | |
| --- | --- |
| Why doesn't honey spoil? | |

| Paraphrases before diversification: | Paraphrases after diversification: |
| --- | --- |
| Why does honey never rot? | Why does honey never rot? |
| Is honey spoiling? | Is honey spoiling? |
| Why does honey not spoil? | Why is honey non-perishable? |
| Why doesn't honey spoil? | Does honey go bad or expire? |

**Cleaning** The first step of this process is to remove any paraphrases that are too similar to others. These were defined as paraphrases with only a few characters or a one-word difference from other paraphrases. Paraphrases were also reviewed for any grammatical or lexical inaccuracies. After that, the intent of the paraphrase is compared to the original CAsT turn. This is to ensure information need is maintained in paraphrases, and there was no deviation from the CAsT turn.

**Diversification** After obtaining clean data, the paraphrases go through a diversification phase. Pre-trained models only output text according to "patterns" learned from crawled or labeled data (*Wallace et al., 2019*). This restricts the creativity and diversity of the automatically generated outputs. Including a human-in-the-loop element injects more lexical diversity into automatically generated paraphrases. To do this, the remaining paraphrases are manually compared against each other by authors. If certain words and expressions are repeated too often, they are substituted using synonyms.

To introduce lexical diversity, commercial search engine result pages (SERP) (*Keyvan & Huang, 2022*) can be used, as they serve as a resource to understand user intent (*Mudrakarta et al., 2018*). Topics in the CAsT dataset are from domains that vary from medical conversations about cancer to ones asking for gardening tips. A method that can allow authors to find more specific words relating to the different topic domains is *via* SERP. To do this, the original CAsT turn was issued in a commercial search engine (Google). The first result page of the search was reviewed by the authors. This facilitates a better understanding of the topic domain and retrieves potential keywords to include for paraphrase diversification. Using this method, words that are very domain-specific, such as "metastasize" and "invasive" for topics about cancer, or "cargo" and "passenger capacity" for topics about airplanes can be added to the paraphrases.

Another resource we used is the "People Also Asked" function available on commercial search engines (*Keyvan & Huang, 2022*). This function displays fully formed queries based on previous searches in the investigated topics. CAsT turns are issued in the Google search engine and the "People Also Asked" questions are reviewed. When appropriate, these questions are included as part of the paraphrase dataset to include more diversity. Table 7 has an example of one turn before and after diversification.

**Feedback and Revealment** In $CAsT^{2021}$, we need to address user feedback and revealments present in some turns. These two types of discourse were added manually by organizers to make topics more conversational. Canonical responses in this dataset are not all relevant

responses to previous turns. In these cases, the organizers included feedback to reflect user dissatisfaction by adding statements such as "What? No, I want to know..." or "That's not what I wanted..." to the start of the subsequent turn. In some cases, the feedback would give hints to the system to continue a certain subtopic such as "No, I meant the funny car. But, that's interesting...". This feedback provides the system with clues on whether it has gone off-topic or if the user wants to shift to a new topic.

Another addition in $CAsT^{2021}$ is user revealments. This was added to a small number of turns in this dataset, and they provide the system with extra information about the user's intent to increase its understanding of information needs. The user would reveal information as part of a turn, such as "I live in Seattle and have a big lawn." or "I'm a runner and I've been feeling tired." These revealments are sometimes needed to interpret later turns in the conversation.

In stage 2 of the paraphrasing solution, feedback and revealments were excluded since the T5-Para model works best with single-turn questions as input due to how it was fine-tuned. To have a correct representation of original information need in the paraphrased conversations, feedback and revealment should be reintroduced and diversified. This was done manually by reintroducing these discourse types but with different expressions while retaining the original intent. Statements such as "What? No, I want to know..." is paraphrased as "That's not what I was looking for." or "You didn't understand me." Revealments such as "I'm a runner and I've been feeling tired." are paraphrased into "I've been feeling tired every time I run." or "I always feel tired when running."

**Final Dataset**    After this data cleaning and diversification stage, the ECAsT dataset is complete and can be used to augment CAsT data and used to challenge CS evaluation robustness.

## Stage 5: Reformulation, retrieval, and re-ranking

In this stage, we use the clean and diverse ECAsT collection to observe the effects of language variation on CS evaluation. The CS system under investigation is a three-stage reformulation, retrieval, and re-ranking pipeline. This approach is a common CAsT solution used by many of the participating teams (*Dalton, Xiong & Callan, 2021*). Many participants relied on this multi-stage approach, and it has been proven effective for the CAsT problem (*Dalton, Xiong & Callan, 2021*).

Different teams implemented different pipeline variations, but they generally started with a pre-trained transformer model for query reformulation. This could be a fine-tuned BERT, T5, or GPT-2 model. This is followed by a retrieval stage that can be either a traditional BM25 system or, in some cases, a dense retrieval system. Then the retrieved passages go through one or more neural re-ranking phases.

To reflect a generalized version of this pipeline, we use the system illustrated in Fig. 5. This starts with a T5-CQR model fine-tuned on CANARD dataset (*Elgohary, Peskov & Boyd-Graber, 2019*), followed by a BM25 retrieval system (*Robertson, Zaragoza et al., 2009*), and ending with a single monoT5 re-ranking model (*Nogueira et al., 2020*). MonoT5 re-ranker receives an input of query and passage pairs that are scored based on their

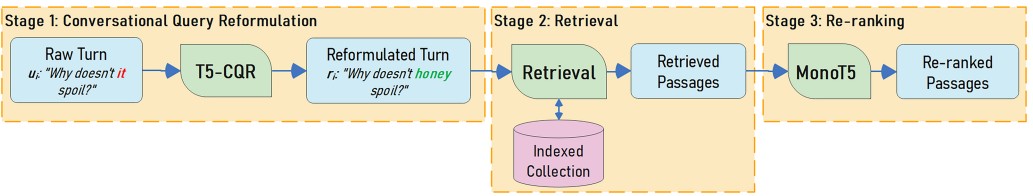

**Figure 5** Reformulation, retrieval, and re-ranking pipeline used to access evaluation metrics robustness to paraphrases.

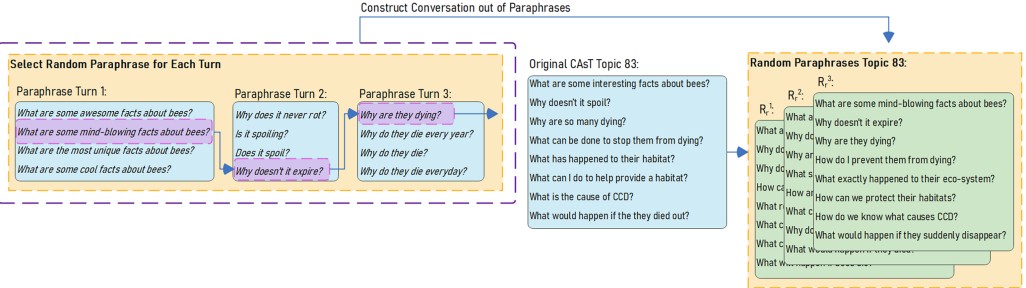

**Figure 6** Selecting random raw paraphrases to create $R_r^1$, $R_r^2$, $R_r^3$ test sets.

relevance. Passages are then re-ordered according to their relevance. Using this system, we retrieve the top 1,000 passages for both manual and raw paraphrases.

To assess the robustness of the reformulation-retrieval-reranking pipeline to language diversity, we create experimental test sets using ECAsT. The goal is to create test sets with the same information need as the original CAsT conversations but with different expressions of information need. Since we have multiple paraphrases for each turn in CAsT, we can construct new conversations by randomly selecting a paraphrase at each turn and adding it to the conversation history. Using this method, we can build multiple test sets with the same information need as the original CAsT conversation.

Figure 6 illustrates how random paraphrases are selected and strung together to construct a different version of topic 83 in $CAsT^{2020}$. Using this method, we create three random tests for each CAsT dataset: $R_r^1$, $R_r^2$, $R_r^3$ using raw paraphrases, and $R_m^1$, $R_m^2$, $R_m^3$ using manual paraphrases. Raw paraphrase test sets go through the three stages of reformulation-retrieval-reranking, but manual paraphrases only go through retrieval and then re-ranking since they are context-independent and do not require reformulation.

TREC CAsT evaluation metrics rely on recall, mean average precision, and Normalized Discounted Cumulative Gain (NDCG). *NDCG@k* is a metric applied in information retrieval that considers both the relevance and rank of passages for each query. Cumulative Gain is the sum of graded relevance scores of all passages in a list. Discounted Cumulative Gain adds a penalty if highly relevant passages appear too far down a list. Normalization scales the metric since some queries are harder than others and produce lower Discounted Cumulative Gain scores. NDCG can be calculated at different depths of rank. The main

| Table 8 | CAsT datasets statistics. | |
|---|---|---|
| Dataset | Topics | Turns |
| CAsT 2019 dev | 30 | 269 |
| CAsT 2019 eval | 50 | 479 |
| CAsT 2020 eval | 25 | 216 |
| CAsT 2021 eval | 26 | 239 |
| Total | 131 | 1,203 |
| ECAsT | 131 | 9,214 |

metric used to rank participant submissions by TREC CAsT is NDCG@3. This is to focus on high-precision and quality responses in the top three ranks.

To study the effects of language variation, we analyze NDCG@3 score for the paraphrase test sets compared to the original CAsT baseline. TREC evaluation is heavily dependent on assessed relevance judgment files (QREL). QREL is a list of scored passages. We explore whether QREL incompleteness affects NDCG@3 scores of paraphrase test sets and how well it reflects the quality of returned passages. We expect language diversity introduced by paraphrases will return unjudged passages.

## EXPERIMENTAL SETUP

In this section, the datasets and experiment setup are detailed. We first present the datasets used for paraphrasing and the passage collection used for retrieval. We also list the different tools and libraries used for pre-processing and indexing. We then clarify neural model hyper-parameters, and evaluation metrics, and we explain the baseline setup.

**Datasets** The CAsT dataset is made up of three releases; $CAsT^{2019}$, $CAsT^{2020}$, and $CAsT^{2021}$. For $CAsT^{2019}$, we have 30 topics included in the development (dev) set and 50 topics in the evaluation (eval) set with an average conversation depth of 9.5 turns. For $CAsT^{2020}$, there are 25 topics with a shorter average depth of 8.6 turns. The final set is $CAsT^{2021}$, which has 26 topics with an average depth of 9.2 turns. All topics in the datasets are open-domain, complex, diverse, and answerable using the collection used for retrieval. All three releases were augmented using paraphrasing. After paraphrasing, the CAsT dataset was augmented by over 665% creating the new ECAsT dataset with 9,214 turns with an average of 7.6 paraphrases per original CAsT turn. To test retrieval and evaluation robustness, only the eval datasets were used, since there is no QREL file available for $CAsT^{2019}$ dev set. Table 8 contains some statistics regarding the datasets used.

**Passage Collection** For passage retrieval, we use MS MARCO (*Nguyen et al., 2016*) and Wikipedia Complex Answer Retrieval (CAR) corpora (*Dietz et al., 2017*) for $CAsT^{2019}$ and $CAsT^{2020}$. After the release of $CAsT^{2021}$, the collection changed to a document-based collection. This is to allow for more complex discourses for the topic set. The collections are MS MARCO documents (*Nguyen et al., 2016*), updated Wikipedia from KILT (*Petroni et al., 2020*), and Washington Post V4 (*Bondarenko et al., 2018*). Documents are split into passages of at most 250 words.

**Evaluation Metrics**   Different metrics are used to evaluate different stages of the solution. First, we present metrics used to evaluate paraphrases and then the metrics used to assess evaluation robustness.

Evaluation of paraphrases is conducted using a combination of human evaluation and automatic evaluation. As discussed earlier, it is difficult to accurately measure the quality of paraphrasing solely relying on automatic measures because we cannot process meaning using such measures (*Niu et al., 2020*). For this reason, paraphrases were assessed using both human evaluation measures (presented in stage 3 of the methodology) and BLEU score as an automatic measure for a comprehensive evaluation.

BLEU (*Papineni et al., 2002*) is the most frequently used measure for paraphrase evaluation (*Zhou & Bhat, 2021*). BLEU score measures the lexical similarity using n-gram overlaps between test and reference sentences. Reference is the ground truth or ideal output, while the test is what is being compared to the reference. A BLEU score of 1 indicates an exact match between the test and reference sentences. A low BLEU score indicates high dissimilarity. It is widely used for many text-to-text transformation tasks, such as translation. This metric is used to measure the quality of the T5-CR model at generating context-dependent turns and the diversity of paraphrases before and after paraphrase diversification.

To evaluate passage retrieval, we use the same performance metrics used for TREC CAsT (*Dalton, Xiong & Callan, 2021*). Retrieval performance is measured using Recall@1000 for $CAsT^{2019}$ and $CAsT^{2020}$, and Recall@500 for $CAsT^{2021}$. Because of the shift from passages to documents, 500 is used instead of the usual 1000 for that year. NDCG@3 and MAP are also used with NDCG@3 as the main metric.

**Tools and Libraries**   There were many steps before and after paraphrasing to prepare and clean data. The Natural Language Toolkit (NLTK) (https://www.nltk.org/) was used for natural language processing, such as sentence segmentation and part-of-speech tagging. The Valence Aware Dictionary and sEntiment Reasoner (VADER) (https://github.com/cjhutto/vaderSentiment) was used for sentiment analysis. To get BLEU score, multi-bleu-detok.perl (https://github.com/EdinburghNLP/nematus/tree/master/data) was used for paraphrase automatic evaluation.

The Anserini toolkit (https://github.com/castorini/anserini) was used for indexing the passage collections and retrieval. Spacy toolkit (https://spacy.io/) was used to segment the $CAsT^{2021}$ document collection into passages. The BM25 retrieval model (*Robertson, Zaragoza et al., 2009*) was used to retrieve the top 1000 passages from the appropriate collection for each CAsT release. For the later re-ranking phase, the 3B-parameter monoT5 re-ranker was used with the settings proposed by *Nogueira et al. (2020)*. The re-rankers were trained with a constant learning rate of 0.001 for 100k iterations. The re-ranking models are available in the PyGaggle (https://github.com/castorini/pygaggle) neural re-ranking library.

**Hyper-Parameter Settings**   We use multiple neural models to accurately paraphrase multi-turn conversations in CAsT. The first is the T5-CQR model built using T5-large and trained with hyper-parameters proposed by *Lin et al. (2020)*. For fine-tuning this model, the CANARD dataset was pre-processed using the same setup in *Elgohary, Peskov & Boyd-Graber (2019)*. As model input, all historical turns and the raw turn were concatenated

with a special separator token between the conversation history and raw turn. The manual turn was set as the model output. T5-CQR is fine-tuned with a constant learning rate of 1e−3 for 4k iterations. T5-large is built using pre-trained weights with over 770 million parameters.

The T5-Para model was initialized with pre-trained weights using the T5-base. It is fine-tuned with a constant learning rate of 3e−4 for 6 epochs. The QQP dataset was processed to only included duplicate questions as training data. T5-CR was built similarly with T5-base, fine-tuned with a constant learning rate of 3e−4 for 6 epochs. The CANARD dataset was pre-processed as training data, with manual turns as input and raw turns as output. T5-base models are built on top of pre-trained weights with 220 million parameters.

None of the inputs for all trained models needed to be truncated. For T5-CQR, a single Google Cloud Platform TPU v3-8 was used to train the T5-large model. For both T5-Para and T5-CR, Google Colab was used for training with a GPU runtime setting since both models are trained using T5-base.

**Baselines**    Multiple systems are being evaluated, each with different baselines to compare its performance against. First, we will present the baselines used to evaluate T5-CR. Then, we discuss the baselines used to measure how well human-in-the-loop improved diversification. Both these experiments rely on BLEU score and are used as an automatic measure for paraphrase evaluation. Lastly, we present the baselines used for CS evaluation robustness testing. This is a retrieval experiment, and we will describe the different datasets and systems used for this experiment.

To evaluate T5-CR, we will compare it to three baselines. The ideal output of T5-CR is the raw turn, since this model's aim is to remove context from manual turns by including appropriate omissions and references. One baseline is to compare manual turns to raw turns in CAsT, denoted as "Original Manual". This reflects the model's starting point, and how similar is the input (manual turns) to the ideal desired output (raw turns). The next baseline is manually removing context from manual turns using rewrites by the first author of this article, denoted as "Rewrite". A final baseline is a rule-based approach that uses the NLTK part-of-speech tagging to automatically remove nouns and proper nouns from the manual turn, denoted as "Entity Removal".

To measure paraphrase diversity before and after human-in-the-loop intervention, we measure the BLEU score of the paraphrases compared to source turns used for paraphrase generation before author diversification. The lower the BLEU score, the more diverse the paraphrase is. We do the same and measure BLEU score after human-in-the-loop diversification. This will measure how well authors introduced more language variation into paraphrases.

To test retrieval using the new paraphrase dataset and its effect on CS evaluation, we use the original CAsT turns as baselines. For raw turn baseline, denoted as $CAsT_r^{year}$, CAsT turns are reformulated before retrieval and re-ranking. The $CAsT_m^{year}$ baseline is where CAsT manual turns are used for retrieval and then re-ranking since turns are context-independent. These are compared with their corresponding random paraphrase test sets $R_r^1$, $R_r^2$, $R_r^3$ and $R_m^1$, $R_m^2$, $R_m^3$. The random paraphrase test sets were built with

unique paraphrases in each set. All retrieval baselines and test sets go through the same reformulation-retrieval-reranking pipeline.

# EXPERIMENTAL EVALUATION AND RESULTS

In this section, the experiment results will be presented for all systems in order to address the RQs in the following manner:

- **RQ1:** *How can we employ paraphrasing to augment multi-turn conversation datasets using multiple neural models?*
  To answer RQ1, we analyze whether the proposed multi-stage paraphrasing solution produced accurate and diverse paraphrases. First, we present the results of the T5-CR system evaluation and how well it removes context from manual turns and replaces it with appropriate references and omissions. After that, human evaluation for paraphrases is presented to evaluate the accuracy and diversity of the generated paraphrases used to augment CAsT data.

- **RQ2:** *How well can we paraphrase context-dependent turns compared to context-independent turns?*
  We answer RQ2 by comparing the paraphrases generated using reformulated turns *versus* manual turns to identify whether one source produces better paraphrases according to human evaluation.

- **RQ3:** *Can automatic paraphrase generation and human-in-the-loop improve paraphrase quality and diversity?*
  We measure the diversity of paraphrases before and after human-in-the-loop intervention to answer RQ3 and see whether human-in-the-loop increase language diversity.

- **RQ4:** *Is the TREC CAsT evaluation metric sensitive to language variation via paraphrasing? How is the metric affected by incomplete relevance judgments?*
  We focus on RQ4 by assessing the robustness of CS evaluation and its bias to language diversity using the paraphrase test sets. First, we put our randomly generated paraphrase sets into the general CAsT solution of reformulation-retrieval-reranking. We investigate how the paraphrases affect the metrics compared to the original CAsT baselines. We then examine retrieved passages by analyzing how many returned passages are scored using the official CAsT QREL files. Finally, we manually judge new unjudged passages (passages not already judged by CAsT organizers in QREL), add them to QREL, and explore how the metrics are affected by this addition and the sensitivity to incomplete relevance judgments.

## T5 for context removal (RQ1)

Here we investigate the performance of T5-CR and how well it identifies turn context and replaces it with appropriate pronouns, references, or omissions. We use both $CAsT^{2020}$ and $CAsT^{2021}$ to test this system. The baselines and system outputs are compared with raw turns as our ground truth (reference file). BLEU score takes into account only lexical similarity and not meaning. Meaning is essential to this solution; however, BLEU still indicates how well T5-CR generates raw turns.

**Table 9  T5-context removal performance versus baselines.**

| | BLEU score | |
| --- | --- | --- |
| | $CAsT^{2020}$ | $CAsT^{2021}$ |
| Original manual | 44.72 | 44.92 |
| Rewrite | **56.57** | **56.01** |
| Entity removal | 30.16 | 34.09 |
| T5-CR | 53.70 | 51.85 |

The "Original Manual" baseline tells us the starting similarity between raw and manual turns. Manual and raw have the same information need; the only difference is raw turns contain omissions and references in place of context. So, we expect them to have a certain degree of similarity initially. For example, the manual turn "Why doesn't honey spoil?" is very similar to the raw turn "Why doesn't it spoil?". The "Rewrite" baseline is our human-labeled data; we expect this to be the best-performing system as authors manually edit the manual turns to remove context. The last baseline, "Entity Removal", is the rule-based approach to context removal using part-of-speech tagging. We use well-known NLP techniques to tag and remove proper nouns and nouns as a simplified context removal solution (*Srinivasa-Desikan, 2018*). Table 9 displays the baselines BLEU score along with T5-CR performance.

The starting similarity between manual and raw turns is a BLEU score of 44.72 for $CAsT^{2020}$ and 44.92 for $CAsT^{2021}$. The best-performing system with the highest BLEU score is the human "Rewrite". However, even with manual context removal, we can see that only a score of 56.57 and 56.01 is achieved. By nature, this type of linguistic task is very subjective. We can express a turn with varying omissions and references in many ways. We can see that, predictably, this system is the best performing for both $CAsT^{2020}$ and $CAsT^{2021}$. The "Entity Removal" baseline is the worst performing system with a BLEU score of 30.16 and 34.09 for $CAsT^{2020}$ and $CAsT^{2021}$, respectively. Instead of bringing the manual turns closer to the desired raw turns, it made them more dissimilar. T5-CR achieves a score close to a human "Rewrite" with scores of 53.70 and 51.85 for $CAsT^{2020}$ and $CAsT^{2021}$, respectively. This is a good score, given the limitations of BLEU and the subjectivity of the task.

## Paraphrase evaluation (RQ1 & RQ2)

Human evaluation was conducted on a sample of 2,550 paraphrases with five annotators for each task. To measure inter-rater agreement, Fleiss kappa was used. *Fleiss kappa* is a well-known multi-rater generalization of Cohen's kappa that measures agreement between two or more raters assigning categorical observations (*Fleiss, 1971*). Initially, Fleiss kappa was very low for all measures ($k$ was between 0.07 and 0.12 indicating "slight agreement" *Landis & Koch, 1977*). This could be due to the subjectivity of this task. Annotations had to be cleaned to reach an acceptable Fleiss kappa score.

The measure with the lowest Fleiss kappa was "Conversational Entailment" ($k = 0.07$). To select a subset of paraphrases with adequate agreement, we choose all paraphrases that have at least 40% agreement between annotators. After that, annotator ratings were

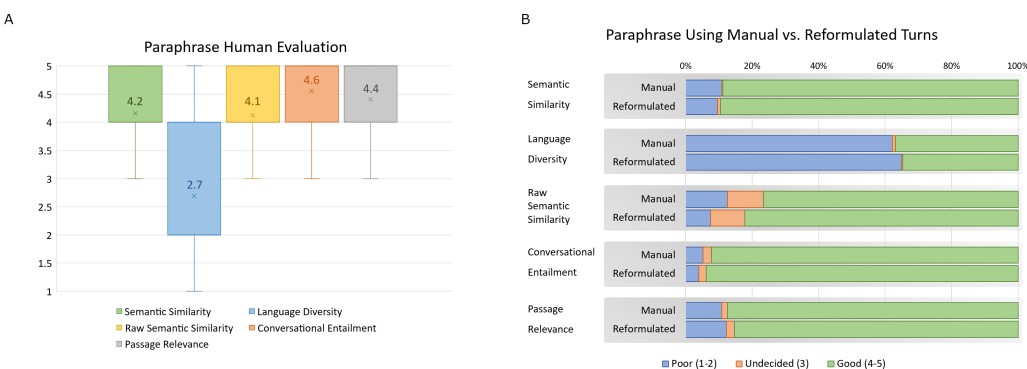

**Figure 7** (A–B) Paraphrase human evaluation results for all measures with agreement between three or more annotators.

grouped into three categories: ratings of 1 and 2 were grouped as "poor" paraphrases, 3 as "undecided", and 4 and 5 as "good" paraphrases. Fleiss kappa works on categorical ratings where each category is independent of all other categories. However, in this scenario, categories are not unrelated. A rating of 1 or 2 generally agrees that the paraphrase is poor, and a rating of 4 or 5 generally agrees that it is good.

By selecting paraphrases with an agreement of 40% and higher amongst annotators and grouping the ratings into three categories, Fleiss kappa is now between 0.26 and 0.36, which indicates "fair agreement" (*Landis & Koch, 1977*). This gives us a subset of 1090 paraphrases. We found that an acceptable score given Fleiss kappa tends to have very low kappa values even in cases of strong agreement between annotators. This is due to how the statistic tends to assume lower values of agreement than expected (*Falotico & Quatto, 2015*). The paraphrases were then labeled with a score between 1 to 5 according to an agreement between three or more annotators for each measure under investigation. The measure labels are summarized in Fig. 7A.

Boxplots in Fig. 7A illustrate the distribution of all measures along with their average values. Most measures score relatively well with "Semantic Similarity", "Raw Semantic Similarity", "Conversational Entailment" and "Passage Relevance" averaging 4.2, 4.1, 4.6, and 4.4 respectively. This indicates that paraphrases maintain the same meaning as the original CAsT turn for both the manual and raw versions of the paraphrase. It also indicates that the new paraphrases fit well into conversation history and retain the same relevant passage as the original CAsT turn.

"Language Diversity" scored the lowest rating with an average of 2.7. This shows that paraphrases are too similar to original CAsT turn in language and expressions. This could be due to the limitation of the neural model. This weakness in diversity was addressed with human-in-the-loop intervention in the paraphrasing cleaning and diversification stage. Language diversity is essential for quality paraphrases.

In Fig. 7B, we can see the measures according to the type of input during paraphrase generation. Turns paraphrased with reformulated turns are measured against turns paraphrased using manual turns. Measures labeled 1 or 2 are grouped as "poor", 3 as

**Table 10  BLEU score before and after diversification step.**

|  | BLEU score |
|---|---|
| $Para_{Auto}$ | 32.06 |
| $Para_{Diverse}$ | 22.60 |

"undecided", and 4 or 5 as "good". We can see that they generally scored similarly across all measures. The difference is not very apparent. This shows that in the cases where manual turns are not available, such as with automatically collected conversations from online resources, reformulation still achieves good paraphrases. Since manual turns require human intervention and are more expensive to collect, this provides another solution to paraphrase generation for multi-turn conversations.

### Paraphrase diversity (RQ3)

In this section, we use BLEU to measure the lexical dissimilarity of paraphrases before and after human-in-the-loop intervention (*Zhou & Bhat, 2021*). Language diversity was the lowest-performing measure during human evaluation. To compensate for this weakness, all paraphrases went through a cleaning and diversification step. We can measure the success of this step by using BLEU to measure how different the paraphrases are from the original turn before and after this manual diversification.

A lower BLEU score indicates more diversity between the reference and test sentence (*Chen & Dolan, 2011*). The ground truth reference sentences are the source turns used as input to T5-Para. This reference will be used against two tests. One test is the T5-Para generated paraphrases without any human intervention denoted as $Para_{Auto}$. The other test is the same paraphrases but after human-in-the-loop diversification, denoted as $Para_{Diverse}$. The full paraphrase dataset is used with turns for all CAsT releases. The scores are reported in Table 10.

As can be noted in Table 10, automatically generated paraphrases are quite different than the source turns with a BLEU score of 32.06. This indicates that T5-Para did produce diverse paraphrases. However, it is important to introduce as much diversity as possible since according to human evaluation, annotators still saw similarities between the two. After diversification, BLEU score went down by 29.5% to 22.60 indicating an increase in language diversity. This shows that manual diversification was successful at improving dissimilarity between the original CAsT turn and the paraphrase.

### Assessing evaluation robustness *via* paraphrases (RQ4)

To assess evaluation robustness to language diversity, we run raw and manual paraphrase test sets through the same retrieval pipeline as the CAsT baselines. All test sets have the same information need as their original counterpart, the only difference is in the language expression. If the evaluation is robust, we would not see a major drop in performance. In Table 11, the metrics of both manual and raw paraphrases are displayed.

The main metric under investigation is NDCG@3 since this is what is used to rank CAsT submissions. As we can note in Table 11, there is a significant drop in NDCG@3 across all paraphrase test sets compared to corresponding CAsT baselines. Manual turns are

**Table 11  Paraphrase test sets versus CAsT baselines performance after reformulation, retrieval, and re-ranking.**

| | Manual turns | | | | Raw turns | | | |
|---|---|---|---|---|---|---|---|---|
| | $R_m^1$ | $R_m^2$ | $R_m^3$ | $CAsT_m$ | $R_r^1$ | $R_r^2$ | $R_r^3$ | $CAsT_r$ |
| | $CAsT^{2019}$ | | | | | | | |
| NDCG@3 | 0.506 | 0.512 | 0.500 | **0.624** | 0.449 | 0.470 | 0.450 | **0.571** |
| Recall@1000 | 0.675 | 0.668 | 0.640 | 0.785 | 0.606 | 0.601 | 0.607 | 0.737 |
| MAP | 0.310 | 0.307 | 0.294 | 0.385 | 0.269 | 0.269 | 0.269 | 0.352 |
| | $CAsT^{2020}$ | | | | | | | |
| NDCG@3 | 0.490 | 0.467 | 0.458 | **0.600** | 0.368 | 0.388 | 0.386 | **0.480** |
| Recall@1000 | 0.620 | 0.631 | 0.623 | 0.730 | 0.473 | 0.528 | 0.528 | 0.588 |
| MAP | 0.311 | 0.301 | 0.299 | 0.408 | 0.219 | 0.236 | 0.236 | 0.309 |
| | $CAsT^{2021}$ | | | | | | | |
| NDCG@3 | 0.539 | 0.566 | 0.553 | **0.600** | 0.369 | 0.374 | 0.340 | **0.384** |
| Recall@500 | 0.660 | 0.665 | 0.665 | 0.692 | 0.517 | 0.483 | 0.527 | 0.514 |
| MAP | 0.340 | 0.349 | 0.348 | 0.405 | 0.236 | 0.217 | 0.226 | 0.248 |

context-independent and consequently have higher NDCG@3 than raw turns. NDCG@3 for manual $CAsT^{2019}$ and $CAsT^{2020}$ dropped on average 12% and 13% from baseline respectively, while raw paraphrases of the same years had an average drop of 12% and 10%, respectively.

$CAsT^{2021}$ had a smaller decrease in NDCG@3 for both manual and raw paraphrases. Manual tests drop an average of 5% while raw tests an average of 2%. $CAsT^{2021}$ shifted from a passage-based to a document-based collection. However, during passage assessment, it was discovered that different versions of SpaCy resulted in inconsistent passage-ids during document segmentation. Due to this error, TREC converted the intended passage-level assessment into a document-level assessment. Results for $CAsT^{2021}$ might not accurately reflect performance. It is unclear if paraphrase test sets retrieved lower quality passage segments than baseline since passage-ids were discarded and only document relevance is available.

## Unjudged passage analysis (RQ4)

To properly investigate the drop in NDCG@3, we manually analyze passages at rank depth 3. We assess whether the drop in NDCG@3 is due to the quality of returned passages or incomplete relevance judgments. For passage analysis, we focus on $CAsT^{2020}$. This is because it contains more complex and harder-to-resolve conversations than $CAsT^{2019}$, and $CAsT^{2021}$ had the passage segmentation error.

We examined the top three passages for 208 out of the total 216 turns in $CAsT^{2020}$. 8 turns were dropped from assessment because they return fewer than three relevant passages. Table 12 summarizes the relevance scores of passages available in the QREL file for 208 turns in $CAsT^{2020}$.

For each test set and baseline, a total of 624 passages were analyzed. Table 13 shows the percentage breakdown of returned passages. Manual paraphrases returned an average of 17% unjudged passages, while manual baseline returned only 2% unjudged passages. For

**Table 12  CAsT 2020 QREL relevance judgement statistics.**

| Relevance Judgement (QREL) | | |
|---|---|---|
| 0 -not relevant | 33781 | 84% |
| 1- Slightly meets | 2697 | 7% |
| 2- Moderately meets | 1834 | 5% |
| 3- highly meets | 1408 | 3% |
| 4-Fully meets | 731 | 2% |
| Total | 40451 | |

**Table 13  CAsT 2020 top 3 passage and unjudged passages score breakdown.**

| | Manual turns | | | | Raw turns | | | |
|---|---|---|---|---|---|---|---|---|
| | $R_m^1$ | $R_m^2$ | $R_m^3$ | $CAsT_m^{2020}$ | $R_r^1$ | $R_r^2$ | $R_r^3$ | $CAsT_r^{2020}$ |
| Judged Passages | 81% | 82% | 84% | 98% | 74% | 75% | 79% | 98% |
| Unjudged Passages | 19% | 18% | 16% | 2% | 26% | 25% | 21% | 2% |
| | Unjudged passages relevance score | | | | | | | |
| 0- Not Relevant | 50% | 57% | 45% | 33% | 73% | 76% | 63% | 90% |
| 1- Slightly Meets | 14% | 17% | 15% | 0% | 9% | 15% | 16% | 0% |
| 2- Moderately Meets | 14% | 14% | 15% | 33% | 7% | 6% | 9% | 10% |
| 3- highly Meets | 8% | 4% | 14% | 33% | 7% | 4% | 13% | 0% |
| 4- Fully Meets | 13% | 8% | 11% | 0% | 6% | 2% | 3% | 0% |

raw paraphrases, an average of 24% of passages were unjudged while the baseline returned 2% unjudged. As we can see, introducing language variation *via* paraphrases returned many new passages not included in the QREL file. Most of the new passages are unique across all test sets as well (85% unique passages for manual paraphrases, and 76% unique passages using raw paraphrases).

To verify the relevance of the passages, they were scored in the same way TREC CAsT performs their passage assessment after pooling (guidelines detailed in Table 6). When scoring the passages, authors used the original CAsT manual turn as a reference to ensure information need is expressed fully. Passages are scored one topic at a time across all test sets to make sure conversation context is retained throughout the scoring process. The new passages' relevance distribution is in Table 13.

As presented in Table 13, an average of 51% of new passages retrieved using manual paraphrases were not relevant while an average of 11% scored a high *4*. On the other hand, raw paraphrases returned an average of 71% not relevant passages and 4% relevant passages with a score of *4*. The manual paraphrases have the advantage of always having the correct information need, so they had a bigger opportunity of returning relevant passages. In the original QREL file released by CAsT, only 2% of the passages are scored *4* (Table 12). We can see the paraphrases successfully returned new high-scoring passages.

**Table 14  CAsT 2020 paraphrase performance before and after addition of new judged passages.**

| | Manual turns | | | | Raw turns | | | |
|---|---|---|---|---|---|---|---|---|
| | $R_m^1$ | $R_m^2$ | $R_m^3$ | $CAsT_m$ | $R_r^1$ | $R_r^2$ | $R_r^3$ | $CAsT_r$ |
| Before new QREL addition | | | | | | | | |
| NDCG@3 | 0.490 | 0.467 | 0.458 | **0.600** | 0.368 | 0.388 | 0.386 | **0.480** |
| Recall@1000 | 0.620 | 0.631 | 0.623 | 0.730 | 0.473 | 0.528 | 0.528 | 0.588 |
| MAP | 0.311 | 0.301 | 0.299 | 0.408 | 0.219 | 0.236 | 0.236 | 0.309 |
| After new QREL addition | | | | | | | | |
| NDCG@3 | 0.541 | 0.496 | 0.503 | **0.598** | 0.389 | 0.400 | 0.400 | **0.476** |
| Recall@1000 | 0.622 | 0.631 | 0.625 | 0.725 | 0.471 | 0.525 | 0.525 | 0.579 |
| MAP | 0.323 | 0.311 | 0.310 | 0.408 | 0.224 | 0.240 | 0.241 | 0.306 |

**New QREL effect on performance (RQ4)**

In this section, we want to measure the effects of the newly scored passages added to QREL. Table 14 displayed the score of $CAsT^{2020}$ before and after the additions of the new passages to QREL.

As displayed in Table 14, NDCG@3 went up for manual test sets by an average of 4% and an average of 2% for raw paraphrases. Original baselines went down slightly, this is due to the effect of adding new relevant passages to QREL. Recall also went down for these baselines while it went up for the paraphrase sets. This shows the sensitivity of CAsT evaluation to systems returning lexical variant answers regardless of their actual relevance. Since CAsT ranks the participating systems based on these scores, this could lead to a ranking bias against systems that add language diversity. This also shows how sensitive NDCG@3 is to the incompleteness of relevance judgments. Recall changed as well while MAP is more stable.

## DISCUSSION AND IMPLICATIONS

The goal of this study was to create a novel and larger conversational paraphrase dataset based solely on CAsT datasets to augment the available conversational data and also to assess evaluation robustness and bias. This research addressed an interesting and novel problem, as there are no paraphrasing solutions for multi-turn conversations to date. Traditionally neural approaches can not handle missing context in input turns.

There are two major parts to the study; in the first part, we explored how to create a novel multi-turn conversation paraphrase dataset, called Expanded-CAsT (ECAsT). After that creation, we take ECAsT and use it to investigate the robustness of CAsT evaluation and the evaluation's bias to language diversity and expression. Using paraphrasing, we were able to augment CAsT data by 665%, and we make ECAsT publicly available for CS research advancement.

To create the dataset, we explored RQ1 and how to paraphrase multi-turn conversations using a combination of methods. First, we do this by creating a multi-stage solution that can paraphrase context-dependent turns. The paraphrases are then diversified using a human-in-the-loop approach. Lexical substitutions are introduced using SERP and the

"People Also Asked" function is used to retrieve new paraphrases of the CAsT turns, as well.

The multi-stage solution first reintroduces context into turns to reformulate them into context-independent turns that can be then used to generate paraphrases. In order to remove context again after paraphrasing, a novel model T5-CR is trained and used to simulate raw turns. This is a necessary step as without this model, paraphrases will look like single-turn questions and not part of a multi-turn conversation. Evaluation of this solution is done using a combination of human and automatic evaluation. The evaluation showed how well the system performed for multi-turn paraphrase generation. The multi-stage solution performed well on all measures, it only fell slightly on the diversity of paraphrases. By following this multi-stage solution, we were able to successfully create a diverse multi-turn paraphrase dataset.

To explore how well the solution paraphrases raw context-dependent turns, we compare the human evaluation results of these turns *versus* paraphrases of manual turns. RQ2 shows that the performance of paraphrases generated from raw turns is comparable to paraphrases from manual turns. This means that the multi-stage paraphrasing solution can be used on datasets that do not have manually labeled turns. For RQ3, we include a human-in-the-loop approach to paraphrase generation to improve language diversity and ensure the quality of paraphrases. Using BLEU score, we show that human-in-the-loop intervention was able to increase paraphrase diversity by 29.5%. This improved the paraphrases' language diversity and allowed for more creative paraphrases that will add more value for many applications and uses.

The second part of this study was to use ECAsT to understand potential weaknesses in CAsT evaluation. CS evaluation is a major topic of investigation in information search and retrieval research domains. This problem introduces a variety of new challenges due to the complexities of such systems. CAsT aims to create a benchmark dataset and evaluation for CS. However, traditional offline evaluations do not address many of the challenges inherent in CS. We focus on exploring one major weakness of this evaluation by introducing language diversity *via* paraphrases in ECAsT. Using this dataset, we answer RQ4 by challenging the robustness of evaluation and bias to new language diversity. We run retrieval experiments using the paraphrases by randomly constructing conversations that have the same information need as CAsT conversation but with different expressions.

Our experiments show that CAsT evaluation is biased towards paraphrases due to incomplete relevance judgments. Unjudged passages returned by paraphrases are not assessed by organizers and are considered not relevant. This means using this evaluation approach will unfairly rate systems that might actually be more robust to language diversity. We also show that language diversity in conversations returns more diverse unique passages. Including these new passages into relevance judgments changed NDCG@3 scores of experiments and showed how sensitive this metric is to incomplete judgments. This is a major flaw in the existing evaluation, as by nature CS systems need to be robust to language diversity to improve user satisfaction. Users will always have a variety of ways of expressing information needs. NDCG@3 is affected by missing passages and as the main metric to score CAsT submission would penalize systems that account for language diversity.

There are many other implications that can be concluded concerning conversational paraphrasing and robustness of evaluation and bias to paraphrases such as:

- With a multi-stage solution and human-in-the-loop techniques, we can paraphrase multi-turn conversations using existing models, such as T5-CQR and T5-Para, by adding a novel new T5-CR model that creates context-dependent turns.
- The multi-stage solution paraphrases context-dependent turns and context-independent turns with comparable quality.
- Including human creativity by using human-in-the-loop approaches improves the quality of automatically generated paraphrases.
- We create ECAsT, a novel conversational paraphrase dataset that is a beneficial contribution to the field due to its numerical size and lexical diversity. ECAsT can be used to augment available datasets, create new interesting models, or for robustness tests.
- The conversation paraphrases was used to test the robustness of CAsT evaluation to language diversity. Results show that evaluation has a negative bias toward language diversity and unfairly measures these systems.
- Using new paraphrases returns a larger and more diverse pool of unjudged passages. CAsT would benefit by including paraphrases in their challenge as it would allow for more diverse passages ranked high in retrieval to the assessment pool.
- NDCG@3 score proved very sensitive to incomplete relevance judgments. This would make CS systems that are more robust to language diversity score lower based on this main metric.

## CONCLUSION AND FUTURE WORK

Conversational search and its applications introduce a variety of new challenges and limitations due to the novelty of the field. These systems need to be able to understand the missing context and user information needs regardless of conversation length and user expression. TREC CAsT addresses many of its challenges by aiming to create a CS benchmark. However, the datasets used were very small, and their evaluation is restricted to traditional offline evaluation. We address data limitations by building ECAsT, a novel multi-turn conversation paraphrase dataset. ECAsT was built with the CAsT turns as the original resource using a novel multi-stage solution that uses both existing models such as T5-CQR and T5-Para, and by introducing novel T5-CR to complete the solution. We also use human-in-the-loop techniques, such as SERP, to include more lexical substitutions, and the "People Also Asked" function to get new complete paraphrases. This new dataset augments CAsT data by 665%, has many applications, and is a valuable contribution to the field.

Paraphrases have many applications in conversation passage retrieval. We use this paraphrase dataset to assess the robustness of CAsT evaluation and identify its bias towards language diversity. Experiments revealed that language diversity is unjustly scored due to incomplete relevance judgments. We also explore the benefits of adding language diversity in improving the collection of pooled passages for CAsT assessment.

A major strength of this research is demonstrating how paraphrasing is used to augment the limited data in CAsT to create the ECAsT larger dataset that can be used to build and test large-scale neural models. This type of automatic data augmentation is easier to use than manually collecting a large dataset, as it needs less human intervention which is naturally expensive. However, one weakness is the need to use a multi-stage solution that requires many neural models, since there are no multi-turn paraphrasing models and datasets available. This requires fine-tuning and running many pre-trained models that are computationally expensive.

There are many future work directions that can be explored using the newly created and publically available ECAsT. One interesting future work is to create a one-step multi-turn paraphrasing solution using ECAsT by fine-tuning a new neural model. This can be compared with the multi-stage solution presented here. Another potential future research work is to explore new evaluation solutions other than these offline metrics that can handle language diversity better than traditional TREC evaluation.

### Funding
The authors received no funding for this work.

### Competing Interests
The authors declare there are no competing interests.

### Author Contributions
- Haya Al-Thani conceived and designed the experiments, performed the experiments, analyzed the data, performed the computation work, prepared figures and/or tables, authored or reviewed drafts of the article, and approved the final draft.
- Bernard J Jansen conceived and designed the experiments, analyzed the data, authored or reviewed drafts of the article, and approved the final draft.
- Tamer Elsayed conceived and designed the experiments, authored or reviewed drafts of the article, and approved the final draft.

### Data Availability
   The codes and dataset are available in the Supplemental Files.

### Supplemental Information
Supplemental information for this article can be found online at http://dx.doi.org/10.7717/peerj-cs.1328#supplemental-information.

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
