# Peer review of "ECAsT: a large dataset for conversational search and an evaluation of metric robustness"

_PeerJ Computer Science, doi:10.7717/peerj-cs.1328_

## Round 0.1 · original submission · Minor Revisions

Please Revise your paper according to the reviewer's comments.
Thank you very much.

·

Basic reporting

The English level of the article is satisfying. Occasionally, some phrases need to be rewritten (to eliminate redundancy) e.g. "The meaning and diversity of paraphrases is evaluated with human evaluations conducted through crowd-sourcing and automatic evaluation."
References are sufficient. SOA is good.
Figures are readable, tables are well explained.
The study hypothesis are presented adequately, and the results show the merit of the research.

Experimental design

The aim of the research is congruent with the purpose of the journal. The only issue I find here is the use of mTurk. I would have first tried to gather data via voluntaries. However, my guess is that your approach doesn't influence at all the results of the study. Nevertheless, I encourage you to further emphasize the use of Amazon mTurk over other means. Another issue is with the ethical consent of the humans involved in any part of the research process. I was wondering if you have requested this prior to their actual involvement.

Validity of the findings

The data in the files supplied is relevant. Conclusions are well written.

Reviewer 2 ·

Basic reporting

This paper provide a new dataset called Expanded-CAst (ECAsT) for The Text REtrieval Conference Conversational assistance track. According to the authors, the ECAst is 665% more extensive in terms of size and language diversity than the CAsT dataset. The ECAsT contains more than 9200 turns. Also, the author mentioned that they use ECAsT to assess the robustness of traditional metrics for conversational evaluation used in CAsT and identify its bias toward language diversity. The authors claims that introducing language diversity via paraphrases in the ECAsT dataset returned up to 24% new passages compared to only 2% using CAsT baseline.

In general, I think this paper is a little bit ambiguous and unclear to me. I did not fully understand how the ECAsT dataset is generated and what are the additional text/semantic resources of the dataset: Does the ECAsT totally comes from expanding the CAsT dataset using rules? The author may want to discuss more about the resources for generating the new dataset.

The references are sufficient in this paper. The background introduction to ECAsT is fairly detailed.

The paper is drafted in a routine structure: Introduction, literature review, methodology, experiments and discussion. Enough details are provided on how to evaluate the proposed new dataset using different metrics.

In general, the authors should provide more details on how the new dataset ECAsT is built: What are the additional text/semantic resources for generating ECAsT (resources beyond CAsT)? But taking the evaluation results using the metrics, I believe that the new dataset is indeed more extensive in terms of size and language diversity comparing to ECAsT.

So, I will recommend minor revision to this paper: Please provide more details on how the new dataset ECAsT is built: What are the text/semantic resources of the dataset?

Experimental design

The authors build the new dataset ECAsT based on the existing benchmark dataset CAsT. Roughly speaking, there are five stages:

Stage 1: Conversational Query Reformulation.
Stage 2: Paraphrasing CAsT Dataset.
Stage 3: Human Evaluation (evaluate generated paraphrases through crowd-sourcing human annotators).
Stage 4: Data Cleaning and Diversification.
Stage 5: Reformulation, Retrieval, and Re-Ranking.

Enough details are provided in each stage. However, after reading through the five stages, I still cannot figure out what are the text/semantic resources for generating ECAsT, except the initial CAsT dataset? The authors should briefly discuss this issue.

Validity of the findings

The authors provide detailed description on how to evaluate the proposed dataset with respect to context removal, paraphrase evaluation, paraphrase diversity, un-judged passage analysis and new QREL effect on performance. Results show that the ECAsT out-performs the CAsT dataset according to the proposed metrics.

Therefore, I think in general the dataset is valuable.

Additional comments

No additional comments.

---

## Round 0.2 · accepted · Accept

I recommend it for publication.

---

## Author Rebuttal · Round 0.2

**PeerJ Computer Science Journal**

**Manuscript Number:** #80890

**Title:** ECAsT: a large dataset for conversational search and an evaluation of metric robustness

This response letter contains our replies to the reviewers' suggestions and comments. The reviewer comments are in **bold**, and we include our responses in green with excerpts from the manuscript in *Italic*. Under each reviewer's comment, we reference the positions in the manuscript that specifically address the comments with appropriate excerpts. We specify the excerpts location on the page by specifying the page and paragraph number (i.e., Page 1, Para 2) We also highlighted the corresponding changes in the manuscript.

We believe we have addressed the reviewers' comments in the manuscript's current version. We thank the reviewers for their observations and constructive comments that have greatly improved the research presented in this version of the manuscript.

## Senior Editor Comments

**It is my opinion as the Academic Editor for your article - ECAsT: a large dataset for conversational search and an evaluation of metric robustness - that it requires a number of Minor Revisions.**
**My suggested changes and reviewer comments are shown below and on your article 'Overview' screen.**
**Please address these changes and resubmit. Although not a hard deadline please try to submit your revision within the next 10 days.**

We thank the editor for the opportunity to revise this manuscript. We believe we have addressed the reviewers' comments in the manuscript's current version.

## Reviewer #1 Comments:

### *Basic reporting*

**The English level of the article is satisfying. Occasionally, some phrases need to be rewritten (to eliminate redundancy) e.g. "The meaning and diversity of paraphrases is evaluated with human evaluations conducted through crowd-sourcing and automatic evaluation."**

Thanks for the note. We have revised the manuscript and removed the quoted phrase, as well as other redundant phrases. Additionally, we have copy-edited this version of the manuscript.

The above redundant phrase was reworded in the "Abstract" to:
> Page 1, Para 1: *"The meaning and diversity of paraphrases are evaluated with human and automatic evaluation."*

**References are sufficient. SOA is good.**

Thanks for the positive comment.

**Figures are readable, tables are well explained.**

Thanks for the positive comment.

**The study hypotheses are presented adequately, and the results show the merit of the research.**

Thanks for the positive comment.

*Experimental design*

**The aim of the research is congruent with the purpose of the journal. The only issue I find here is the use of mTurk. I would have first tried to gather data via voluntaries. However, my guess is that your approach doesn't influence at all the results of the study. Nevertheless, I encourage you to further emphasize the use of Amazon mTurk over other means. Another issue is with the ethical consent of the humans involved in any part of the research process. I was wondering if you have requested this prior to their actual involvement.**

Thank you for the constructive criticism. We acknowledge that mTurk has its weaknesses in some cases and that volunteers can sometimes be the better option. However, due to the size of the evaluation dataset (12,750 evaluation tasks), we made the decision to use mTurk as the more efficient approach. The task under investigation also doesn't require specialized understanding from the annotators other than a proficiency in the English language. For those reasons, we decided to use mTurk versus other annotator options. The annotators also were notified as being part of a research study in the mTurk task description.

This justification was clarified in this version of the manuscript in "Section 3: Methodology" under the sub-section "Stage 3: Human Evaluation":

> Page 8, Para 3: *"Amazon mTurk was used instead of recruiting local volunteers due to the size of the evaluation task (12,750 tasks). The evaluation task also does not require any domain-specific knowledge from annotators other than a general proficiency in the English language. Annotators were informed beforehand that the task is part of a research study."*

### *Validity of the findings*

**The data in the files supplied is relevant. Conclusions are well written.**

Thank you for the constructive comments. We believe this version of the manuscript is much improved based on the received comments and suggestions.

### Reviewer #2 Comments:

**Basic reporting**

**In general, I think this paper is a little bit ambiguous and unclear to me. I did not fully understand how the ECAsT dataset is generated and what are the additional text/semantic resources of the dataset: Does the ECAsT totally comes from expanding the CAsT dataset using rules? The author may want to discuss more about the resources for generating the new dataset.**

Thank you for your comments. ECAsT comes from expanding CAsT dataset by training a paraphrase generation model, using it to generate paraphrases, then manually refining and diversifying the automatically-generated paraphrases. Therefore, CAsT is the only textual resource, as all the paraphrases are generated using turns from this dataset. We have also exploited commercial search engine tools, such as the search engine results pages and the "People Also Ask" feature in the search engine. No other resources were used.

This is highlighted in the manuscript in "Section 1: Introduction":

> Page 2, Para 3: "*In this research, we create a novel conversation paraphrase dataset that is both larger and more diverse than existing datasets. We make the dataset, called Expanded-CAsT (ECAsT), publicly available for the advancement of CS research. Here, we present how ECAsT is built from only the TREC CAsT datasets as a resource using both neural models and human-in-the-loop diversification. ECAsT is constructed with CAsT turns using automatic*

*paraphrase generation and commercial search engine tools, such as  the search engine results pages and the "People Also Ask" feature in the search engine."*

Some more details were clarified and highlighted in "Section 3: Methodology":

Page 6, Para 1:*"CAsT turns are automatically paraphrased using a pre-trained transformer-based multi-stage solution followed by human-in-the-loop techniques using search engine results page and the ``People Also Asked" feature."*

Page 6, Para 1:*"Stages 1 to 4 describe how ECAsT is built using TREC CAsT as a resource."*

Page 10, Para 3: *"To introduce lexical diversity, commercial search engine result pages (SERP) (Keyvan and Huang, 2022) can be used, as they serve as a resource to understand user intent (Mudrakarta et al., 2018). Topics in the CAsT dataset are from domains that vary from medical conversations about cancer to ones asking for gardening tips. A method that can allow authors to find more specific words relating to the different topic domains is via SERP. To do this, the original CAsT turn was issued in a commercial search engine (Google). The first result page of the search was reviewed by the authors. This facilitates a better understanding of the topic domain and retrieves potential keywords to include for paraphrase diversification."*

Page 10, Para 4: *"Another resource we used is the "People Also Asked"' function available on commercial search engines (Keyvan and Huang, 2022). This function displays fully formed queries based on previous searches in the investigated topics. CAsT turns are issued in the Google search engine and the "People Also Asked" questions are reviewed. When appropriate, these questions are included as part of the paraphrase dataset to include more diversity."*

**The references are sufficient in this paper.**

Thanks for the positive comment.

**The background introduction to ECAsT is fairly detailed.**

Thanks for the positive comment.

The paper is drafted in a routine structure: Introduction, literature review, methodology, experiments and discussion. Enough details are provided on how to evaluate the proposed new dataset using different metrics.

Thanks for the positive comment.

In general, the authors should provide more details on how the new dataset ECAsT is built: What are the additional text/semantic resources for generating ECAsT (resources beyond CAsT)? But taking the evaluation results using the metrics, I believe that the new dataset is indeed more extensive in terms of size and language diversity comparing to ECAsT.

Thank you for your comments. We believe this version of the manuscript states the resources more clearly based on the received comments and changes to the manuscript, as detailed in response to your first comment above.

So, I will recommend minor revision to this paper: Please provide more details on how the new dataset ECAsT is built: What are the text/semantic resources of the dataset?

Thank you for your comments. We believe this version of the manuscript states the resources more clearly based on the received comments and changes to the manuscript, as detailed in response to your first comment above.

*Experimental design*

The authors build the new dataset ECAsT based on the existing benchmark dataset CAsT. Roughly speaking, there are five stages:

Stage 1: Conversational Query Reformulation.

Stage 2: Paraphrasing CAsT Dataset.

Stage 3: Human Evaluation (evaluate generated paraphrases through crowd-sourcing human annotators).

Stage 4: Data Cleaning and Diversification.

Stage 5: Reformulation, Retrieval, and Re-Ranking.

Enough details are provided in each stage. However, after reading through the five stages, I still cannot figure out what are the text/semantic resources for

**generating ECAsT, except the initial CAsT dataset? The authors should briefly discuss this issue.**

Thank you for your comments. We believe this observation is addressed in the previous comments and corresponding changes to the manuscript.

*Validity of the findings*

**The authors provide detailed description on how to evaluate the proposed dataset with respect to context removal, paraphrase evaluation, paraphrase diversity, un-judged passage analysis and new QREL effect on performance. Results show that the ECAsT out-performs the CAsT dataset according to the proposed metrics.**

Thanks for the positive comment.

**Therefore, I think in general the dataset is valuable.**

Thanks for the positive comment and the support of the research presented in this manuscript.